# Experimental facility had a greater effect on growth performance, gut microbiome, and metabolome in weaned pigs than feeding diets containing subtherapeutic levels of antibiotics: A case study

**Michaela P. Trudeau[1][¤], Wes Mosher[2], Huyen Tran[3], Brenda de Rodas[3], Theodore P. Karnezos[3], Pedro E. Urriola[1], Andres Gomez[1], Milena Saqui-Salces[1], Chi Chen[1,2], Gerald C. Shurson[1] ***

1 Department of Animal Science, University of Minnesota, St. Paul, Minnesota, United States of America, 2 Department of Food Science and Nutrition, University of Minnesota, St. Paul, Minnesota, United States of America, 3 Purina Animal Nutrition, Gray Summit, Missouri, United States of America

¤ Current address: Vita Plus Corporation, Madison, Wisconsin, United States of America

* shurs001@umn.edu

**Data Availability Statement:** The 16S rRNA sequence data analyzed in this manuscript are

## Abstract

The objective of this study was to define changes in the intestinal metabolome and microbiome associated with growth performance of weaned pigs fed subtherapeutic concentrations of antibiotics. Three experiments with the same antibiotic treatments were conducted on the same research farm but in two different facilities (nursery and wean-finish) using pigs weaned at 20-days of age from the same source herd and genotype, and fed the same diets formulated without antibiotics (NC) or with 0.01% chlortetracycline and 0.01% sulfamethazine (AB). Pigs were weighed and feed disappearance was determined on days (d) 10, 21, and 42 post-weaning to calculate average daily gain (ADG), average daily feed intake (ADFI), and gain:feed (G:F). On d 42, one pig/pen was selected for blood and ileal and cecal content collection. Targeted and untargeted metabolomic profiles were determined in serum and cecal contents using liquid chromatography-mass spectrometry, and composition of bacterial communities in intestinal content samples was determined by sequencing the V4 region of the 16s rRNA gene. Metabolomics and microbiome data were analyzed using diverse multivariate and machine learning methods. Pigs fed AB had significantly greater ($P$ < 0.05) overall ADG and ADFI compared with those fed NC, and pig body weight, ADG, and G:F were also significantly different ($P$ < 0.05) between experiments. Differences ($P$ < 0.05) in serum metabolome along with ileal and cecal microbiome beta diversity were observed between experiments, but there were no differences in microbiome alpha diversity between experiments or treatments. Bacteria from the families Clostridiaceae, Streptomycetaceae, Peptostreptomycetaceae, and Leuconostocaceae were significant biomarkers for the AB treatment. In addition, pigs fed AB had increased serum arginine, histidine, lysine, and phenylalanine concentrations compared with NC. Percentage error from a random forest analysis indicated that most of the variation (8% error) in the microbiome was explained by the

available in NCBI Sequence Read Archive under accession BioProject ID PRJNA950003. The link is: https://dataview.ncbi.nlm.nih.gov/object/PRJNA950003?reviewer=qc9rohdoa3u4huurtn063r90il.

**Funding:** Partial research funding for conducting animal experiments was provided by Land O' Lakes/Purina and managed by BdR. Additional funding provided by Land O' Lakes/Purina to the University of Minnesota was managed by GCS for metabolomics and microbiome analysis and Research Assistantship support for MPT. BdR, TK, and HT from Land O' Lakes/Purina were involved in study design, data collection and analysis, decision to publish, and preparation of the manuscript.

**Competing interests:** The authors have declared that no competing interests exist.

facility where the experiments were conducted. These results indicate that facility had a greater effect on growth performance, metabolome, and microbiome responses than feeding diets containing subtherapeutic levels of antibiotics.

## Introduction

The efficiency of pork production has improved over time as a result of genetic improvements, feed formulation practices, and feed additives that improve nutritional efficiency [1–6]. Historically, subtherapeutic concentrations of antibiotics as growth promoters (AGPs) in swine diets were commonly used for improving health and growth performance, especially under poor sanitary housing conditions [1, 7, 8]. However, this practice has contributed toward the development of antibiotic-resistant bacteria, which is a significant human health risk [9–13]. These concerns have led to a global need to reduce antibiotic use in both humans and livestock [10]. In the United States, government regulations have been implemented to restrict use of AGPs in animal diets through the Veterinary Feed Directive [14]. As a result, there is tremendous interest in the global feed industry to identify and use feed additives that are as effective as AGPs for improving animal growth and health.

Unfortunately, little is known about the mechanisms associated with growth promotion responses of AGPs when added to swine diets. Understanding these mechanisms may be useful for identifying feed additives that provide similar responses as alternatives to AGPs. Growth promotion responses from adding AGPs to swine diets are inconsistent [7, 15, 16]. Differences in growth performance responses from feeding AGPs among studies has been attributed to differences in health status of pigs, with disease challenged pigs having a greater improvement in growth performance than healthy pigs [2, 15, 16]. A potential explanation for the differences in response to antibiotics among experiments is that there are differences in microbial communities among sections (ileum and cecum) of the pig gastrointestinal tract. The ileum is the main site for nutrient absorption and the cecum has the greatest concentration of bacteria, which makes them ideal sites for further exploration of the effects of AB on the swine microbiome. Therefore, the intestinal microbiome has been shown to vary between different experiments [17] and likely determines if growth performance from feeding AGPs are observed as a result of significant differences at the community and functional levels. In addition, because AGPs suppress microbial communities and their metabolites, more dietary energy is partitioned toward growth rather than using energy for maintaining immune tolerance associated with the microbiome [18]. However, no studies have been conducted to compare potential differences in growth, metabolome, and gut microbiome responses of weaned pigs from the same source herd and fed the same diets containing AGPs in different facilities.

Therefore, we hypothesized that changes in the ileum and cecum microbiome and the cecum and serum metabolome from feeding diets containing common AGPs can be associated with the magnitude of improvement in growth performance of in weaned pigs. Therefore, the objectives of this study were to determine growth performance responses, characterize metabolome changes, and determine the intestinal bacterial community composition in weaned pigs fed diets with or without AGPs.

## Materials and methods

The feeding experiments and sample collections were conducted at the Purina Animal Nutrition Research Farm (Gray Summit, MO, USA) using standard operating protocols. All

metabolome and microbiome laboratory analyses, as well as associated data and statistical analyses were conducted at the University of Minnesota (St. Paul, MN, USA).

## Animals, housing, and experimental design

Experiment (exp) 1 occurred from June 11 to July 26, 2018; exp 2 was conducted from August 20 to October 4, 2018; and exp 3 was performed from July 16 to August 30, 2018. Experiments 1 and 2 were completed in the same environmentally controlled nursery facility, while exp 3 was conducted in an environmentally controlled wean-to-finish facility located on the same research farm. The nursery facility included plastic flooring, five-hole plastic feeders, and nipple waterers. The wean-to-finish facility included concrete slatted flooring, five-hole metal feeders, and cup waterers. In both facilities, room temperature was maintained at about 30°C during the first week after weaning, and then decreased by 1.5°C per week during each 6-week exp. Before each exp, facilities were completely washed and sanitized using the same chemicals and standard operating procedures established by Purina Animal Nutrition.

The minimum number of replicates for each exp was determined by calculating the necessary sample size (n) to achieve statistical significance at $P \leq 0.05$ and power of 0.80. Results from previous studies focused on microbiome and metabolomic analyses were used as inputs to G*Power 3.1 (Kiel University, Germany), which indicated that 8 replicates were required to detect significant differences. Experiment 1 and 3 utilized eight replicates per treatment, while exp 2 included nine replicates per treatment. A completely randomized block design was used in each exp, and pigs were blocked by initial body weight (BW) and sex.

All pigs were from the same genetic line (PIC Camborough × PIC 337, Hendersonville, TN, USA), and were obtained from the same farm and sow unit (Purina Animal Nutrition). Pigs were weaned at 20-d of age and had an average initial BW of 6.5 kg in each exp. All pigs were vaccinated for *Streptococcus suis* and *Mycoplasma hyorhinis* (Autogenous Bacterin, Philbro Animal Health, Teaneck, NJ) at 5 to 7-d of age and received a booster inoculation at weaning. Pigs were also vaccinated for *Haemophilus parasuis* (ParaSail, Newport Laboratories, Worthington, MN), and *Salmonella* Typhimurium (Enterisol-Salmonella T/C; BI, St. Joseph, MO) 7-d prior to weaning; and Circovirus Type 2 (Fostera PCV Chimera, Zoetis, Charles City, IA) at weaning. Throughout each exp, pigs were monitored for health status and any medication treatments used were recorded. In exp 1, two pigs were removed from the AB treatment because of a *Streptococcus suis* infection, and one pig was removed from the NC group because of lameness. In exp 2, one pig in AB treatment and one pig in the NC treatment were removed because of lameness. In exp 3, one pig was removed from the AB group and one pig from the NC group due to a *Streptococcus suis* infection.

## Dietary treatments

A 3-phase feeding program was used in each exp, which consisted of feeding a phase 1 diet from weaning (d 0) to d 10 post-weaning, a phase 2 diet from d 10 to d 21, and a phase 3 diet from d 21 to d 42 post-weaning. All diets were formulated to exceed the National Research Council (2012) energy and nutrient requirements for nursery pigs [18] and were manufactured in pelleted form at the Purina Research Manufacturing Unit (Gray Summit, MO, USA). Dietary treatments consisted of feeding antibiotic diets (AB) with 0.5% Aureo Mix 10-10S (Zoetis; Charles City, IA) providing 0.01% chlortetracycline and 0.01% sulfamethazine throughout each 42-d exp, and negative control diets (NC) that contained no antibiotics. All pigs had *ad libitum* access to feed and water throughout each exp.

## Statistical analysis of growth performance data

Growth performance data (BW, ADG, ADFI, and G:F) were analyzed for absence of outliers and normal distribution using the UNIVARIATE procedure of SAS (SAS Institute, Cary, NC). Pen was considered the experimental unit and experimental data were analyzed as a randomized complete block design using the GLIMMIX procedure of SAS with time as repeated measure with autoregressive 1 variance structure. Degrees of freedom were calculated using the Kenward-Roger method. Replicate was considered as random effect, while fixed effects included dietary treatments and experiment. The effect of facility was not included as a main effect because it was not an independent variable from the experiment. The main effects included treatment, experiment, and time, along with the interactions of treatment × time, experiment × time, experiment × treatment, and experiment × treatment × time. Significant differences were declared at $P \leq 0.05$.

## Sample collection

For metabolomic and microbiome analyses, one pig with BW closest to the median BW of the pen was selected for blood and intestinal content collection on day 42 of each exp (62 days of age). Blood samples were collected via venipuncture of the jugular vein in Vacutainer® blood collection tubes (BD, Franklin Lakes, NJ, USA) and then centrifuged at $2,000 \times g$ for 15 min at 4°C. Serum was then aliquoted and stored at -80°C. Pigs were subsequently euthanized using the penetrating captive bolt gun followed by exsanguination to collect digesta samples following IACUC protocol SRU-009 for animal handling and health approved by the Purina IACUC committee. After euthanasia, the entire intestinal tract was removed, placed on a sterile surface, and sterilized utensils were used to collect intestinal content samples. Approximately 1.5 mL of cecal contents were collected from the lateral side of the cecum and 1.5 mL of ileum contents were collected 30 cm proximal to the ileocecal junction. Each sample was immediately snap frozen in liquid nitrogen and stored at -80°C.

## Metabolomics analysis

Liquid chromatography–mass spectrometry (LC-MS) based metabolomic analysis involved several steps including sample preparation, chemical derivatization, LC-MS analysis, data deconvolution and processing, multivariate data analysis, and marker characterization and quantification [19, 20]. Deproteinization of serum was conducted by mixing one volume of serum with 19 volumes of 66% aqueous acetonitrile (ACN) followed by centrifugation at $18,000 \times g$ for 10 min at room temperature. Cecal content samples were mixed with 50% aqueous ACN containing 5 μM glycocholic acid-$^{13}C_1$ in 1:10 (w/v) ratio. Samples were sonicated for 10 min followed by mixing using a vortex mixer. After mixing, samples were centrifuged at $18,000 \times g$ for 10 min and the supernatant was collected to obtain cecal sample extracts.

For detection of metabolites containing amino groups in their structure, samples were derivatized with dansyl chloride (DC) prior to the LC-MS analysis. Briefly, 5 μL of samples or standards were mixed with 5 μL of 100 μM $p$-chlorophenylalanine (internal standard), 50 μL of 10 mM sodium carbonate, and 100 μL of DC (3 mg/mL in acetone). The mixtures were incubated at 25°C for 15 min, centrifuged ($18,000 \times g$) for 10 min, and the supernatant was transferred into high-performance liquid chromatography vials for LC-MS analysis.

Samples were derivatized with 2-hydrazinoquinoline (HQ) prior to the LC-MS analysis for identification of carboxylic acids, aldehydes, and ketones species [21]. Briefly, 2 μL of sample were added into a 100 μL of freshly prepared acetonitrile solution containing 1 mM 2,2′-dipyridyl disulfide (DPDS), 1 mM triphenylphosphine (TPP), and 1 mM HQ. The reaction mixture was incubated at 60°C for 30 min, chilled on ice and then mixed with 100 μL of ice-cold $H_2O$.

After centrifugation at $21,000 \times g$ for 10 min, the supernatant was transferred into a HPLC vial for LC-MS analysis.

A 5 μL aliquot of sample prepared from serum or cecal fluid was injected into an Acquity ultra-performance liquid chromatography (UPLC) system (Waters, Milford, MA) and separated by a gradient of mobile phase ranging from water to 95% aqueous acetonitrile containing 0.1% formic acid over a 10 min run. The LC eluant was introduced into a Xevo-G2-S quadrupole time of flight mass spectrometry (QTOFMS, Waters) for mass measurement and ion counting. Capillary voltage and cone voltage for electrospray ionization was maintained at 3 kV and 30 V for positive-mode detection, or at -3 kV and -35 V for negative-mode detection, respectively. Source temperature and desolvation temperature were set at 120˚C and 350˚C, respectively. Nitrogen was used as both cone gas (50 L/h) and desolvation gas (600 L/h), and argon was used as collision gas. For accurate mass measurement, the mass spectrometer was calibrated with sodium formate solution (range $m/z$ 50–1,000) and monitored by the intermittent injection of the lock mass ($[M+H]^+ = m/z$ 556.2771 and ($[M+H]^- = m/z$ 554.2615) in real time. Mass chromatograms and mass spectral data were acquired and processed using MassLynx[TM] software (Waters, Milford, MA, USA) in centroided format.

After data acquisition in the UPLC-QTOFMS system, chromatographic and spectral data of samples were deconvoluted by MarkerLynx[TM] software (Waters, Milford, MA, USA). A multivariate data matrix containing information on sample identity, ion identity (RT and $m/z$) and ion abundance was generated through centroiding, deisotoping, filtering, peak recognition, and integration. The intensity of each ion was calculated by normalizing the single ion counts (SIC) *versus* the total ion counts (TIC) in the whole chromatogram.

In the untargeted metabolite analysis, the processed data matrix was exported into Rstudio software (Boston, MA, USA) and then analyzed by unsupervised principal components analysis (PCA). The vegan package was used to determine the weighted Bray-Curtis matrix [22]. Major latent variables in the data matrix were determined as the principal components of PCA model.

For the targeted analysis, individual metabolite concentrations were calculated using the ratio between the peak area of metabolite and the peak area of internal standard and fitting with a standard curve using QuanLynx software (Waters, Milford, MA). The concentration of targeted analysis was evaluated for normality using the Shapiro test in Rstudio. An analysis of variance between treatments was completed using the Kruskal-Wallis rank sum test using the stats package in Rstudio. If significant, multiple comparisons were performed using the pgirmess package in Rstudio [23]. Differences of group means were considered significant if $P < 0.05$ and a trend was based on $0.05 \geq P \leq 0.10$.

## Microbiome analysis

All cecal (n = 52) and ileal (n = 52) content samples were submitted for DNA extraction and sequencing of the 16S rRNA bacterial gene (V4 region). The extractions were performed using DNeasy PowerSoil DNA extraction kit (Qiagen, Hilden, Germany) following manufacturers' instructions. Bead beating was performed for 3 min at maximum speed (6 m/s) during DNA extraction. The DNA samples were then submitted to the University of Minnesota Genomics Center where they were processed using their standardized methods. Library prep was completed using a previously described dual-indexing method [24]. Marker gene sequencing was then completed using the Illumina MiSeq Next Generation platform (Illumina, San Diego, CA, USA) with a targeted average sequencing depth of 100,000 reads per sample.

Amplicon Sequence Variants (ASVs) were identified from raw Illumina sequence reads using cutadapt, fastx, and Qiime 2 bioinformatics tools [25–27]. The adapter sequences,

barcodes and primers were removed from the pair-end reads and low-quality reads were discarded. Primers were removed using the default parameters of the cutadapt program. Empty lines were then removed from the output and then the sequences were filtered using a quality score of 50 and all other parameters set to default settings. The paired ends were then merged using bbmap merger and singletons were discarded [28]. Data were then imported into Qiime 2 (version 2019.4), where sequences were demultiplexed and processed through the dada2 pipeline to identify ASVs. Taxonomic analysis of the ASVs obtained was then completed using a pre-trained classifier. After data processing, 72% of the reads were retained from the raw reads.

After obtaining the ASVs, data were transferred into Rstudio (version 3.5.3) where the rarefied richness and Simpson alpha diversity were determined using the vegan package (version 2.4.2) [29]. In addition, the vegan package was used to determine the weighted Bray-Curtis matrix. This matrix was used to calculate beta diversity between treatments. Finally, species indicator analyses were performed using the labdsv package [30]. The indicator value is a product of frequency and abundance of a given taxon across all samples belonging to a specific treatment. When an indicator value is close to 1, the taxon is present in most samples of that treatment and in high abundance compared with other treatment groups. Only indicator values greater than 0.8, for a given bacterial taxon, are reported in this study. Indicator values were reported to identify representative species for AB and NC, and for exp (facility). A random forest analysis was also completed using the random forest package in Rstudio [31]. Confusion matrices and percentage error were reported for comparisons between treatments, between experiments, and between facilities where the exp were conducted.

## Results

### Growth performance

**Antibiotic vs. negative control.** There was no effect of dietary treatment on BW or G:F, but pigs fed AB had greater ($P < 0.05$) ADG and ADFI compared with those fed NC (Table 1). As expected, BW, ADG, and ADFI increased ($P < 0.05$) with time, and there was a time × treatment interaction ($P < 0.05$) for ADG. This interaction indicates that AB was more effective at improving ADG during 21-d to 42-d post-weaning compared with day 0 to 10 and day 10 to 21 post-weaning growth (Table 1). This interaction was only observed for ADG and not for BW, ADFI, or G:F.

**Experiment comparison.** There were no differences in initial BW between the three experiments, but pigs in exp 3 had less ($P < 0.05$) final BW and G:F compared with pigs in exp 1 and 2, which indicates that pigs housed in the wean-finish facility had reduced growth performance compared with pigs housed in the nursery facility (Table 1). There were no interactions of exp × time or exp × treatment, indicating that the effect of exp on BW, ADG, ADFI, or G:F was independent of treatment and time. The interaction between treatment × exp × time was only significant for G:F, indicating that the relationship between gain efficiency in the NC and AB group varied based on the exp and time ($P = 0.047$; Table 1).

### Metabolome

**Antibiotic vs. negative control.** Pigs fed AB had greater ($P < 0.05$) serum concentrations of arginine, histidine, lysine, phenylalanine, and valine compared with those fed NC (Table 2). However, the untargeted analysis, which compared all detectable metabolites, showed no clear separation between the serum metabolome of pigs fed AB compared with those fed NC (Fig 1A). A different pattern of metabolites was observed in the cecal contents compared with the serum metabolome. For the targeted metabolites, there were no differences in short chain fatty

**Table 1. Interactive effects of dietary antibiotics (Trt), experiment (Exp), and time point (Time) on pig growth performance during a 42-day post-weaning feeding period [1].**

| Experiment | 1 | | 2 | | 3 | | SEM[3] | P- value | | | |
|---|---|---|---|---|---|---|---|---|---|---|---|
| Treatment[2] | NC | AB | NC | AB | NC | AB | | Trt | Time | Exp | Trt × Time × Exp |
| *Body weight (kg)* | | | | | | | | | | | |
| Day 0 | 6.69 | 6.68 | 6.58 | 6.58 | 6.19 | 6.20 | 0.73 | 0.20 | $P < 0.05$ | $P < 0.05$ | |
| Day 10 | 8.48 | 8.27 | 8.10 | 7.99 | 7.48 | 7.60 | | | | | 0.97 |
| Day 21 | 13.89 | 13.61 | 12.78 | 13.07 | 12.08 | 12.61 | | | | | |
| Day 42 | 26.86 | 27.26 | 26.22 | 27.26 | 24.24 | 26.40 | | | | | |
| *Average daily gain (kg)* | | | | | | | | | | | |
| Day 10 | 0.18 | 0.16 | 0.15 | 0.14 | 0.13 | 0.14 | 0.02 | $P < 0.05$ | $P < 0.05$ | $P < 0.05$ | 0.93 |
| Day 21 | 0.49 | 0.49 | 0.42 | 0.46 | 0.42 | 0.46 | | | | | |
| Day 42 | 0.62 | 0.65 | 0.64 | 0.68 | 0.58 | 0.65 | | | | | |
| *Average daily feed intake (kg)* | | | | | | | | | | | |
| Day 10 | 0.19 | 0.18 | 0.17 | 0.17 | 0.16 | 0.17 | 0.03 | $P < 0.05$ | $P < 0.05$ | 0.67 | 0.71 |
| Day 21 | 0.56 | 0.54 | 0.49 | 0.51 | 0.49 | 0.53 | | | | | |
| Day 42 | 0.89 | 0.90 | 0.90 | 0.95 | 0.87 | 1.00 | | | | | |
| *Gain:Feed* | | | | | | | | | | | |
| Day 10 | 0.94[a] | 0.88[b] | 0.87 | 0.83 | 0.80 | 0.84 | 0.02 | 0.98 | $P < 0.05$ | $P < 0.05$ | $P < 0.05$ |
| Day 21 | 0.88 | 0.91 | 0.87 | 0.90 | 0.86 | 0.85 | | | | | |
| Day 42 | 0.69 | 0.72 | 0.72 | 0.72 | 0.67 | 0.66 | | | | | |

[1]Data are presented as least squared means. Experimental data were analyzed as a randomized complete block design using the GLIMMIX procedure of SAS with time as repeated measures with autoregressive 1 variance structure. Degrees of freedom were calculated using Kenward-Roger method. Random effects included replicate, while fixed effects included dietary treatments and experiments.

[2]NC = negative control; AB = antibiotic treatment.

[3]SEM = standard error of the mean

acids and bile acid concentrations between pigs fed AB and NC treatments (Table 3). However, there were differences in untargeted metabolomic profiles between AB and NC (Fig 2A), which were primarily due to the presence of antibiotic metabolites in the cecal samples from pigs fed AB.

**Experiment comparison.** There were differences ($P < 0.05$) in metabolite profiles of pigs in exp 1 compared with exp 2 and 3, indicating that the exp had a greater effect on the serum metabolome than dietary treatment (Fig 1B). When evaluating the metabolomes among the 3 experiments, unlike the responses observed in serum metabolome, greater differences in the metabolome were evident when comparing the metabolite profile of exp 3 with exp 1 and 2.

## Microbiome

**Antibiotic vs. negative control.** Feeding AB had no effect on the microbial alpha diversity, which was assessed using the Simpson index, and the rarefied richness of the cecal or ileal contents (Fig 3A and 3B). In addition, there were no differences in bacterial diversity between pigs fed AB and those fed NC in cecal or ileal contents (Fig 4A and 4C).

Despite no apparent impact on alpha and beta diversity of the microbiome when pigs were fed AB, indicator species analyses were used to identify potential microbial indicators of the AB treatment. Microbial indicators may be useful for detecting bacterial species or ASVs that were highly affected by feeding antibiotics. Indicators were only reported if the values were greater than 0.8 and had a different relative abundance between the dietary treatment groups based on Wilcoxon tests. In the cecal samples, there were two unidentified ASVs from the

**Table 2. Effect of dietary antibiotics on serum amino acid concentrations of pigs on day 42 post-weaning.**

| Experiment | 1 | | 2 | | 3 | | Combined | | P-value[2] |
|---|---|---|---|---|---|---|---|---|---|
| Treatment[1] | AB | NC | AB | NC | AB | NC | AB | NC | |
| *Free amino acids, mg/g* | | | | | | | | | |
| Alanine | 0.32 | 0.31 | 0.47 | 0.45 | 0.35 | 0.38 | 0.39 | 0.38 | 0.99 |
| Arginine | 0.53 | 0.46 | 0.60 | 0.51 | 0.52 | 0.46 | 0.55 | 0.48* | 0.01 |
| Asparagine | 0.17 | 0.15 | 0.20 | 0.16 | 0.18 | 0.19 | 0.18 | 0.17 | 0.47 |
| Aspartic acid | 0.01 | 0.01 | 0.01 | 0.01 | 0.02 | 0.01 | 0.01 | 0.01 | 0.87 |
| Citrulline | 0.06 | 0.04 | 0.06 | 0.06 | 0.07 | 0.08 | 0.06 | 0.06 | 0.45 |
| Glutamic acid | 0.28 | 0.25 | 0.27 | 0.36 | 0.46 | 0.43 | 0.33 | 0.35 | 0.73 |
| Glutamine | 1.17 | 1.06 | 1.30 | 1.09 | 1.12 | 1.09 | 1.20 | 1.08 | 0.15 |
| Glycine | 1.14 | 1.06 | 1.31 | 1.20 | 1.21 | 1.13 | 1.22 | 1.13 | 0.15 |
| Histidine | 0.09 | 0.07 | 0.12 | 0.09 | 0.10 | 0.08 | 0.10 | 0.08* | 0.01 |
| Leucine/Isoleucine | 0.36 | 0.31 | 0.43 | 0.39 | 0.37 | 0.35 | 0.39 | 0.35 | 0.17 |
| Lysine | 0.49 | 0.42 | 0.65 | 0.57 | 0.62 | 0.42 | 0.59 | 0.47* | 0.02 |
| Methionine | 0.03 | 0.03 | 0.02 | 0.02 | 0.02 | 0.03 | 0.03 | 0.03 | 0.66 |
| Ornithine | 0.12 | 0.10 | 0.17 | 0.16 | 0.17 | 0.16 | 0.15 | 0.14 | 0.30 |
| Phenylalanine | 0.24 | 0.21 | 0.30 | 0.24 | 0.26 | 0.21 | 0.27 | 0.22* | 0.01 |
| Proline | 0.58 | 0.56 | 0.73 | 0.69 | 0.59 | 0.58 | 0.64 | 0.61 | 0.45 |
| Serine | 0.19 | 0.18 | 0.33 | 0.27 | 0.22 | 0.20 | 0.25 | 0.22 | 0.23 |
| Taurine | 0.06 | 0.03 | 0.28 | 0.16 | 0.04 | 0.03 | 0.13 | 0.08 | 0.11 |
| Threonine | 0.42 | 0.51 | 0.54 | 0.46 | 0.34 | 0.29 | 0.44 | 0.42 | 0.74 |
| Tryptophan | 0.13 | 0.11 | 0.18 | 0.16 | 0.15 | 0.12 | 0.15 | 0.13 | 0.14 |
| Tyrosine | 0.17 | 0.15 | 0.17 | 0.15 | 0.11 | 0.11 | 0.15 | 0.14 | 0.46 |
| Valine | 0.38 | 0.33 | 0.49 | 0.43 | 0.38 | 0.32 | 0.42 | 0.36 | 0.05 |

[1]AB = antibiotic treatment; NC = negative control.

[2]*P*-value from Kruskal Wallis one-way analysis of variance comparing all antibiotic treated pigs to all negative control pigs.

* Values are significantly different ($P < 0.05$) than the negative control within the same experiment.

family *Clostridiaceae* identified as indicators (Table 4). Both ASVs were increased when feeding AB compared with NC. There were no ASVs identified with high indicator values for pigs fed NC. In the ileal samples, three unidentified bacteria ASVs were characteristic of pigs fed AB: *Streptococcaceae* (family) *Streptococcus* (genus), *Clostridiales* (order) *Peptostreptococcaceae* (family), and *Leuconostocaceae* (family) *Weissella* (genus).

**Experiment comparison.** Although no dietary treatment effect was observed, there were differences ($R^2$ = 0.18, $P$ = 0.01) in beta diversity between exp for cecal contents, in a similar manner to changes observed in the cecal metabolome (PERMANOVA $R^2$ = 0.26, $P$ = 0.01; Fig 1B). Microbial diversity differences among exp for ileal contents were significant but only explained 11% of the variation ($R^2$ = 0.11, $P$ = 0.0; Fig 4D).

A random forest analysis was used to compare the ability of the cecal microbiome dataset to predict pig responses to dietary treatment, experiment, or facility. When using the data to predict the dietary treatment, there was a 40% error rate, which indicates that there were not enough differences in the microbiome between dietary treatments to make accurate predictions (Table 5). When this analysis was repeated to make predictions by exp, the error rate was reduced to 28%. Finally, when a random forest analysis was conducted to compare the facility in which the exp was conducted, the error rate was only 8%, indicating that the experiment explains a large portion of the differences in ileal and cecal microbiota and subsequent response to AB.

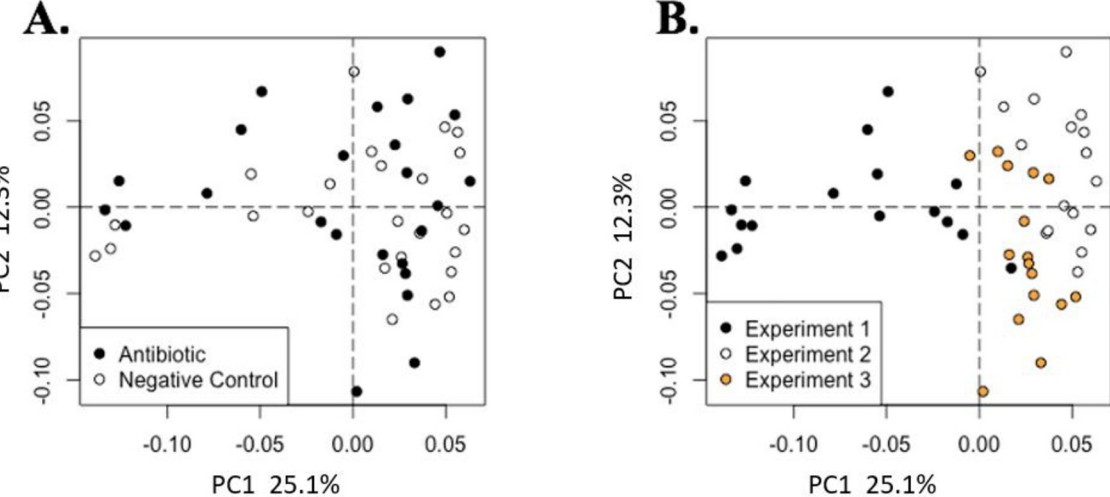

**Fig 1.** Metabolite analysis of serum samples through weighted Bray-Curtis distance ordination of metabolite beta diversity colored by A) treatment (Antibiotic n = 25, Negative Control n = 25) and B) experiment (experiment 1 n = 16, experiment 2 n = 16, experiment 3 n = 18). Lack of separation indicates no major difference in the metabolite diversity among dietary treatments (A; PERMANOVA, $R^2$ = 0.03, $P$ = 0.19), but clear separation by experiment (B; PERMANOVA, $R^2$ = 0.26 $P$ = 0.01), indicating that experiment explained 26% of metabolite variation in this study.

Multiple microbial biomarkers were identified for each facility (Table 6). In the cecal samples, these included unknown ASVs from the families *Veillonellaceae* and *Lachnospiraceae* in the nursery facility, and unknown ASVs from the families *Clostridiaceae* in the wean-finish facility. For ileal content samples, ASVs from the family *Lactobacillaceae* were more abundant in the nursery facility while ASVs from the family *Clostridiaceae* were more abundant in the wean-finish facility. These differences indicate that the facility of the exp can have an impact on the microbial diversity, with differences resulting from only a few variants of bacteria.

## Discussion

One of the many challenges of determining the relative effectiveness of feeding AGPs or other feed additives in swine diets is the lack of reproducibility of responses among experiments [15]. To gain insight on the potential mechanisms of AGPs for improving growth performance, we fed nursery diets without antibiotics (NC) and with antibiotics (AB) in three separate exp for 42 days post weaning to determine and compare growth, metabolic, and microbiome responses in weaned pigs.

Overall, there was an increase in ADG and ADFI when pigs were fed diets containing AB compared with those fed NC, where the increased ADG was a result of enhanced feed intake but there was not an improvement in gain efficiency. These findings are consistent with results from summaries of several experiments that showed inconsistent responses from feeding AGPs to weaned pigs [7, 15, 16]. The interaction between dietary treatment and time for ADG also indicated that pigs responded differently to AB as they increased in age. These findings contradict previous research results which have shown that the growth promotion responses of pigs from feeding antibiotics are greater immediately after weaning compared with subsequent growth stages [1, 16]. We speculate that this greater response to AB in the early post-weaning period is due to a reduction in pathogen pressure on the immune system associated with weaning stress [1, 32]. In our study, the observed lack of growth improvement during the first phase post-weaning in the present study may indicate that pigs experienced minimal

**Table 3. Effect of dietary antibiotics on amino acid, short-chain fatty acid, and bile acid profiles in cecal contents of pigs on day 42 post-weaning.**

| Experiment | 1 | | 2 | | 3 | | Combined | | |
|---|---|---|---|---|---|---|---|---|---|
| Treatment[1] | AB | NC | AB | NC | AB | NC | AB | NC | P-value |
| *Amino acids, mg/g* | | | | | | | | | |
| Alanine | 0.64 | 0.60 | 0.28 | 0.30 | 0.63 | 0.49 | 0.51 | 0.46 | 0.55 |
| Arginine | 0.02 | 0.02 | 0.01 | 0.01 | 0.01 | 0.01 | 0.01 | 0.01 | 0.97 |
| Asparagine | 0.00 | 0.01 | 0.00 | 0.00 | 0.00 | 0.00 | 0.00 | 0.00 | 0.29 |
| Aspartic acid | 0.32 | 0.28 | 0.17 | 0.23 | 0.27 | 0.20 | 0.25 | 0.23 | 0.75 |
| Citrulline | 0.02 | 0.01 | 0.01 | 0.01 | 0.01 | 0.01 | 0.01 | 0.01 | 0.32 |
| Glutamic acid | 1.18 | 0.96 | 0.56 | 0.68 | 0.98 | 0.64 | 0.89 | 0.75 | 0.50 |
| Glutamine | 0.01 | 0.01 | 0.01 | 0.01 | 0.00 | 0.00 | 0.01 | 0.01 | 0.83 |
| Glycine | 0.12 | 0.09 | 0.03 | 0.04 | 0.07 | 0.04 | 0.07 | 0.06 | 0.52 |
| Histidine | 0.01 | 0.01 | 0.01 | 0.01 | 0.01 | 0.01 | 0.01 | 0.01 | 0.80 |
| Leucine | 0.03 | 0.04 | 0.01 | 0.01 | 0.03 | 0.02 | 0.02 | 0.03 | 0.65 |
| Lysine | 0.15 | 0.17 | 0.06 | 0.08 | 0.13 | 0.09 | 0.11 | 0.11 | 0.87 |
| Methionine | 0.01 | 0.01 | 0.00 | 0.01 | 0.01 | 0.01 | 0.01 | 0.01 | 0.31 |
| Ornithine | 0.01 | 0.01 | 0.01 | 0.01 | 0.01 | 0.01 | 0.01 | 0.01 | 0.70 |
| Phenylalanine | 0.05 | 0.07 | 0.02 | 0.03 | 0.05 | 0.04 | 0.04 | 0.04 | 0.78 |
| Proline | 0.05 | 0.06 | 0.03 | 0.03 | 0.04 | 0.04 | 0.04 | 0.04 | 0.66 |
| r-amino-n-butyric acid | 0.01 | 0.01 | 0.01 | 0.01 | 0.01 | 0.01 | 0.01 | 0.01 | 0.67 |
| Serine | 0.02 | 0.03 | 0.01 | 0.01 | 0.02 | 0.01 | 0.01 | 0.02 | 0.53 |
| Taurine | 0.01 | 0.00 | 0.01 | 0.00 | 0.01 | 0.00 | 0.01 | 0.00 | 0.22 |
| Threonine | 0.04 | 0.04 | 0.02 | 0.03 | 0.03 | 0.02 | 0.03 | 0.03 | 0.79 |
| Tryptophan | 0.01 | 0.01 | 0.01 | 0.01 | 0.01 | 0.00 | 0.01 | 0.01 | 0.90 |
| Tyrosine | 0.06 | 0.10 | 0.03 | 0.05 | 0.03 | 0.02 | 0.04 | 0.06 | 0.36 |
| Valine | 0.04 | 0.05 | 0.02 | 0.02 | 0.05 | 0.03 | 0.03 | 0.03 | 0.99 |
| *Short chain fatty acids, mg/g* | | | | | | | | | |
| Acetic acid | 32.44 | 29.60 | 33.38 | 34.22 | 41.22 | 36.13 | 35.59 | 33.35 | 0.48 |
| Propionic acid | 20.97 | 20.78 | 19.99 | 21.88 | 26.91 | 23.72 | 22.52 | 22.12 | 0.87 |
| Butyric acid | 22.15 | 17.41 | 22.23 | 25.04 | 28.48 | 27.28 | 24.20 | 23.32 | 0.72 |
| Valeric acid | 6.05 | 3.67 | 5.33 | 7.54 | 7.69 | 7.42 | 6.31 | 6.26 | 0.96 |
| Isovaleric acid | 57.49 | 124.64 | 22.48 | 66.65 | 32.77 | 19.21 | 36.98 | 70.02 | 0.42 |
| *Bile acids, µg/g* | | | | | | | | | |
| Cholic acid | 0.10 | 0.05 | 0.00 | 0.05 | 0.36 | 0.00 | 0.15 | 0.03 | 0.11 |
| Chenodeoxy cholic acid | 19.48 | 11.88 | 17.84 | 7.98 | 30.03 | 15.46 | 22.27 | 11.62 | 0.29 |
| Lithocholic acid | 111.33 | 101.16 | 59.08 | 138.33 | 151.86 | 151.29 | 105.49 | 130.58 | 0.23 |
| Taurocheno deoxycholic acid | 0.13 | 0.19 | 0.39 | 0.89 | 0.00 | 0.00 | 0.18 | 0.38 | 0.42 |
| Hyodeoxycholic acid | 495.28 | 413.82 | 393.44 | 472.09 | 764.63 | 697.79 | 544.81 | 525.67 | 0.78 |

[1]AB = antibiotic treatment; NC = negative control.

weaning stress or immune challenge, because these factors have been shown to result in significant growth responses from feeding antibiotics [33].

When feeding antibiotics, there were significant increases in the serum amino acid concentrations of lysine, histidine, and phenylalanine. Based on our results, feeding AB increased ADFI by 6.7% which resulted in 0.45 g more lysine consumed per pig per day based on the average dietary lysine content in the overall nursery phase. This increased lysine consumption corresponds to the 25% increase in serum lysine concentrations observed. Results from a previous study showed that a change in serum lysine concentration can be directly related to

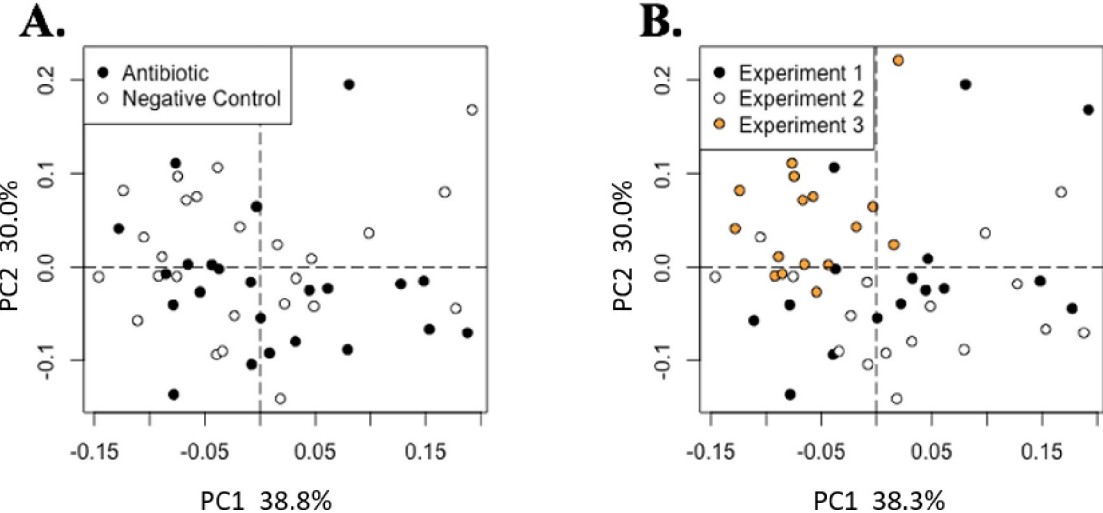

**Fig 2.** Metabolite analysis of cecal samples through weighted Bray-Curtis distance ordination of metabolite beta diversity colored by A) treatment (Antibiotic n = 25, Negative Control n = 25) and B) experiment (experiment 1 n = 16, experiment 2 n = 16, experiment 3 n = 18). Clear separation indicates differences in the cecal metabolome diversity by treatment (PERMANOVA, $R^2$ = 0.04, $P$ = 0.04) and by experiment (PERMANOVA, $R^2$ = 0.16, $P$ = 0.01).

dietary supply of lysine [34]. However, serum lysine concentration is not a measure of absorption, and it must also be considered that lysine catabolism is slower than that of other essential amino acids [35]. Comparatively slower lysine catabolism may explain the greater magnitude of increase in serum lysine concentration than the increase in feed intake. The increased feed intake in pigs fed AB could also explain increased serum levels of histidine because increased serum concentrations of histidine have also been previously reported to have a direct correlation to feed intake [36]. For phenylalanine, the increased concentration in the serum could be reflective of increased health status when pigs are fed an antibiotic. Because of its importance as a component in acute phase proteins, phenylalanine is considered the first limiting amino in health challenged pigs [37]. Therefore, the increased plasma concentrations of phenylalanine observed in the current study may indicate that the pigs fed AB had less inflammation in the absence of a health challenge, but further research is needed to test this hypothesis.

Despite the addition of AGPs to the diet, targeted and untargeted metabolites in the cecal metabolome were unaffected. We hypothesized that pigs fed AB would have differences in bacterial metabolites including short chain fatty acids and bile acids (BA) concentrations in cecal contents, but no differences were observed between pigs fed AB and those fed NC. Results from previous studies have shown that adding antibiotics to pig diets increases secondary BA concentrations in the intestine, and this response has been linked to the mechanism of growth promotion for antimicrobials [38]. The lack of this response in our study may be due to different antibiotics fed (e.g., tylosin and chlortetracycline) in previous studies compared with feeding a combination of chlortetracycline and sulfamethazine in our experiments [38, 39]. Because tylosin and sulfamethazine utilize a different mechanism of action to inhibit bacterial growth, it is likely that they may have a mechanism different from increasing BA concentrations. Another factor that contributed to differences in results between our study and a previous study [38] is that the 16S rRNA gene hypervariable region sequenced between studies (i.e., V1-V2 regions were sequenced in [38], not V4 as in our study). Therefore, the region sequenced may be a third factor contributing to differences among experiments [17].

Despite a lack of dietary treatment differences in alpha and beta diversity of gut microbiome, supplementing AGPs resulted in changes in the proportion of specific bacteria. These

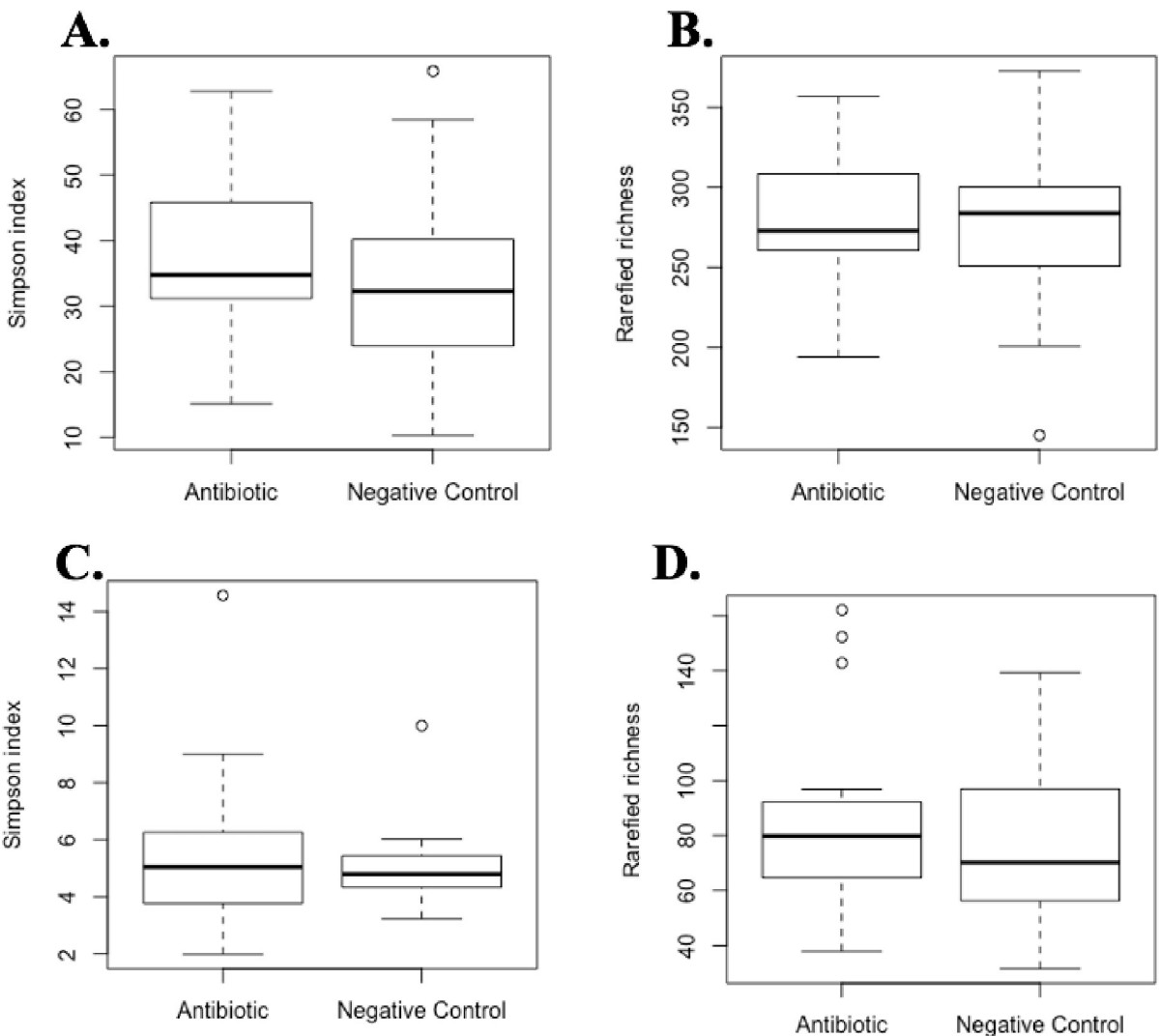

**Fig 3. Microbiome analysis of bacterial composition of cecal and ileal samples.** Treatments include antibiotic treatment and negative control A) The Simpson alpha diversity index for each treatment in cecal content. B) The rarefied richness for each treatment in the cecal content. C) The Simpson alpha diversity index for each treatment in the ileal content. D) The rarefied richness for each treatment in the ileal content. There were no significant differences between the antibiotic and negative control treatment in the Simpson index or rarefied richness measured in cecal and ileal contents ($P > 0.05$).

results are not unusual. Results from other studies have shown that feeding AGPs to pigs caused changes in only *Proteobacteria* and *Escherichia coli* in the fecal microbiome [40]. Similarly, a study that evaluated the effects of feeding antibiotics to pigs on the microbiome in multiple sites of the intestinal tract showed that in the ileum, *Lachnobacterium* was the only genus that contributed to community differences between pigs fed diets without and with antibiotics [41]. In the current study, the abundance of a few bacteria variants changed, with some ASVs identified as indicators for the AB treatment including *Clostridiaceae* (exp 2), *Streptomycetaceae* (exp 3), *Streptococcaceae* (exp 1), *Peptostreptococcacea* (exp 2), and *Leuconostocaceae* (exp 2).

One of the significant findings from this study was the effect of experiment on growth performance, serum and cecal metabolome, and cecal microbiome. The differences in BW, ADG, and G:F responses between exp may be partially due to physical differences (i.e., flooring,

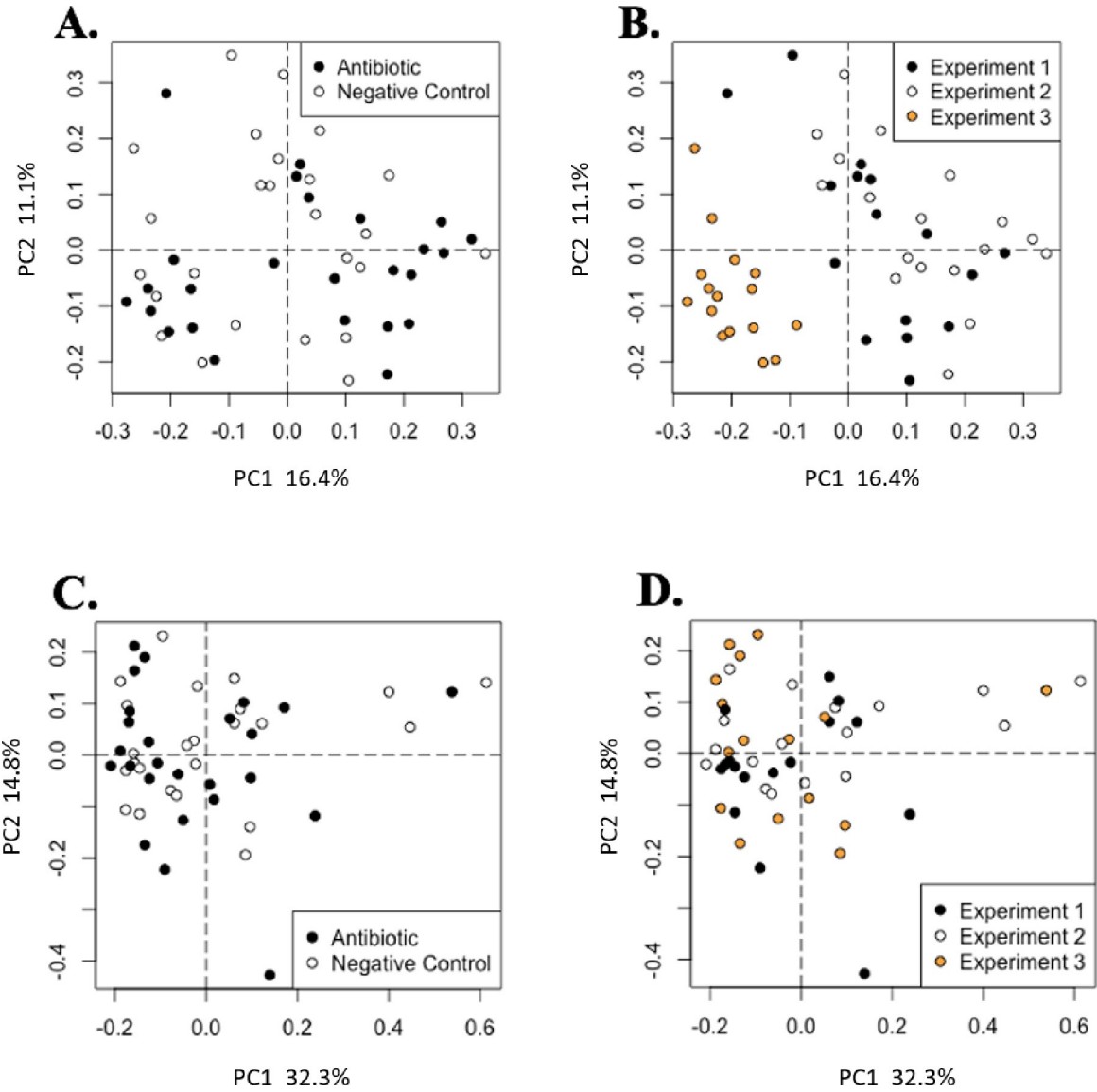

**Fig 4.** Microbiome analysis of cecal and ileal samples through weighted Bray-Curtis distance ordination of microbiome beta diversity in cecal samples colored by A) treatment (Antibiotic n = 25, Negative Control n = 25) and B) experiment (experiment 1 n = 16, experiment 2 n = 16, experiment 3 n = 18). Clear separation indicates differences in the cecal microbiome by experiment (PERMANOVA, $R^2$ = 0.18, $P$ = 0.01) but not by treatment (PERMANOVA, $R^2$ = 0.02, $P$ = 0.19). The weighted Bray-Curtis distance ordination of microbiome beta diversity in ileal samples colored by C) treatment and D) experiment. Clear separation indicates differences in the ileal microbiome by experiment (PERMANOVA, $R^2$ = 0.11, $P$ = 0.0) but not by treatment (PERMANOVA, $R^2$ = 0.0, $P$ = 0.48).

drinkers, feeders) between the wean-finish and nursery facilities where these experiments were conducted, but there may have also been environmental differences between facilities. However, results from a previous study showed that no differences in ADG, ADFI, or G:F were observed for pigs raised in either wean-finish facilities or nursery facilities [42]. The impact of experiment could also be influenced by season, maternal factors, or batch of pigs because each experiment was conducted at a different time during the months from July to October and piglets were obtained from different groups of sows from the same herd. However, if this were the case, the random forest analysis applied on the cecal microbiome data would have resulted in the lower error rate when comparing the effect of dietary treatment in each experiment

**Table 4. Cecal and ileal microbial biomarkers for pigs fed diets without and with antibiotics and mean relative abundance.**

| Cecal content microbial biomarkers | | | | | |
|---|---|---|---|---|---|
| Amplicon Sequence Variant[1] | Experiment | Indicator Value | Negative control (mean[2] ± standard deviation) | Antibiotic (mean[2] ± standard deviation) | P-value[3] |
| k__Bacteria p__Firmicutes c__Clostridia o__Clostridiales f__Clostridiaceae g__SMB53 | 2 | 0.82 | 0.50 ± 0.47 | 2.26 ± 1.61 | 0.01 |
| k__Bacteria p__Actinobacteria c__Actinobacteria o__Actinomycetales f__Streptomycetaceae | 3 | 1.00 | 0 ± 0 | 0.03 ± 0.02 | 0.01 |
| Ileal content microbial biomarkers | | | | | |
| Amplicon Sequence Variant[1] | Experiment | Indicator Value | Negative control (mean[2] ± standard deviation) | Antibiotic (mean[2] ± standard deviation) | P-value[3] |
| k__Bacteria p__Firmicutes c__Bacilli o__Lactobacillales f__Streptococcaceae g__Streptococcus | 1 | 0.85 | 0.17 ± 0.34 | 0.98 ± 0.99 | 0.05 |
| k__Bacteria p__Firmicutes c__Clostridia o__Clostridiales f__Peptostreptococcace | 2 | 0.83 | 0.87 ± 1.06 | 4.36 ± 4.46 | 0.05 |
| k__Bacteria p__Firmicutes c__Bacilli o__Lactobacillales f__Leuconostocaceae g__Weissella | 2 | 0.83 | 0.01 ± 0.04 | 0.06 ± 0.05 | 0.01 |

[1]k represents kingdom, p represents phylum, c represents class, o represents order, f represents family, g represents genus, and s represents species. Bacteria are classified to be as specific as possible with the sequences available.

[2]Means expressed as percentage relative abundance.

[3]P-value from Kruskal Wallis one-way analysis of variance.

separately. Instead, the results from the random forest analysis suggest that the greatest variation in the cecal microbiome is explained by the facility where the exp was conducted, because pigs used in the two experiments conducted in the nursery facility had similar microbial composition.

**Table 5. Percentage error and confusion matrix from random forest analysis in cecal microbiome samples of weaned pigs fed diets without and with antibiotics.**

| Confusion matrix by treatment[1] (error: 40%) | | | | | |
|---|---|---|---|---|---|
| | | Predicted outcome | | | |
| | | NC | AB | Total | |
| True outcome | NC | 9 | 4 | 13 | |
| | AB | 6 | 6 | 12 | |
| | Total | 15 | 10 | 25 | |
| Confusion matrix by experiment (error: 28%) | | | | | |
| | | Predicted outcome | | | |
| | | Experiment 1 | Experiment 2 | Experiment 3 | Total |
| True outcome | Experiment 1 | 8 | 1 | 1 | 10 |
| | Experiment 2 | 4 | 4 | 0 | 8 |
| | Experiment 3 | 1 | 0 | 6 | 7 |
| | Total | 13 | 5 | 7 | 25 |
| Confusion matrix by facility (error: 8%) | | | | | |
| | | Predicted outcome | | | |
| | | Nursery | Wean-Finish | Total | |
| True outcome | Nursery | 18 | 0 | 18 | |
| | Wean-Finish | 2 | 5 | 7 | |
| | Total | 20 | 5 | 25 | |

[1]AB = antibiotic treatment; NC = negative control.

**Table 6. Cecal and ileal microbial biomarkers of each facility and mean relative abundance comparison for among facilities.**

| Cecal samples[1] | | | | | |
|---|---|---|---|---|---|
| Amplicon Sequence Variant[2] | Facility | Indicator value | Nursery (mean[3] ± standard deviation) | Wean-Finish (mean[3] ± standard deviation) | P-value[4] |
| k__Bacteria p__Firmicutes c__Clostridia o__Clostridiales f__Veillonellaceae g__Acidaminococcus | Nursery | 0.93 | 0.40 ± 0.40 | 0.01 ± 0.01 | 0.03 |
| k__Bacteria p__Firmicutes c__Clostridia o__Clostridiales f__Veillonellaceae g__Megasphaera | Nursery | 0.92 | 2.72 ± 1.75 | 0.21 ± 0.34 | 0.01 |
| k__Bacteria p__Firmicutes c__Clostridia o__Clostridiales f__Lachnospiraceae g__Roseburia s__faecis | Nursery | 0.91 | 2.65 ± 2.61 | 0.23 ± 0.23 | 0.02 |
| k__Bacteria p__Firmicutes c__Clostridia o__Clostridiales f__Peptostreptococcaceae | Nursery | 0.86 | 0.29 ± 0.23 | 0.03 ± 0.05 | 0.01 |
| k__Bacteria p__Firmicutes c__Clostridia o__Clostridiales f__Clostridiaceae g__Clostridium s__celatum | Wean-finish | 0.99 | 0.01 ± 0.01 | 0.38 ± 0.36 | 0.01 |
| k__Bacteria p__Firmicutes c__Clostridia o__Clostridiales f__Clostridiaceae g__Clostridium s__butyricum | Wean-finish | 0.96 | 0.01 ± 0.012 | 0.15 ± 0.09 | 0.01 |
| k__Bacteria p__Firmicutes c__Clostridia o__Clostridiales f__Lachnospiraceae g__Lachnospira | Wean-finish | 0.88 | 0 ± 0 | 0.68 ± 0.68 | 0.01 |
| k__Bacteria p__Firmicutes c__Clostridia o__Clostridiales f__Veillonellaceae | Wean-finish | 0.88 | 0 ± 0 | 0.21 ± 0.19 | 0.01 |
| k__Bacteria p__Bacteroidetes c__Bacteroidia o__Bacteroidales f__S24-7 | Wean-finish | 0.88 | 0 ± 0 | 0.02 ± 0.02 | 0.01 |
| k__Bacteria p__Firmicutes c__Clostridia o__Clostridiales | Wean-finish | 0.88 | 0 ± 0 | 0.21 ± 0.26 | 0.01 |
| k__Bacteria p__Bacteroidetes c__Bacteroidia o__Bacteroidales f__S24-7 | Wean-finish | 0.86 | 0.02 ± 0.05 | 0.88 ± 1.25 | 0.01 |
| k__Bacteria p__Firmicutes c__Clostridia o__Clostridiales f__Peptostreptococcaceae | Wean-finish | 0.85 | 0.02 ± 0.04 | 0.09 ± 0.06 | 0.00 |
| k__Bacteria p__Firmicutes c__Clostridia o__Clostridiales f__Lachnospiraceae g__Blautia | Wean-finish | 0.84 | 0.06 ± 0.05 | 0.26 ± 0.32 | 0.01 |
| k__Bacteria p__Bacteroidetes c__Bacteroidia o__Bacteroidales f__Prevotellaceae g__Prevotella s__stercorea | Wean-finish | 0.82 | 0.05 ± 0.08 | 0.19 ± 0.19 | 0.01 |
| Ileal samples[1] | | | | | |
| Amplicon Sequence Variant[2] | Facility | Indicator value | Nursery (mean[3] ± standard deviation) | Wean-Finish (mean[3] ± standard deviation) | P-value[4] |
| k__Bacteria p__Firmicutes c__Bacilli o__Lactobacillales f__Lactobacillaceae g__Lactobacillus | Nursery | 0.81 | 11.43 ± 9.49 | 2.60 ± 3.25 | 0.02 |
| k__Bacteria p__Firmicutes c__Clostridia o__Clostridiales f__Clostridiaceae | Nursery | 0.81 | 0.90 ± 0.73 | 0.21 ± 0.14 | 0.02 |
| k__Bacteria p__Firmicutes c__Clostridia o__Clostridiales f__Clostridiaceae | Wean-finish | 1.00 | 0 ± 0 | 0.37 ± 0.08 | 0.00 |
| k__Bacteria p__Firmicutes c__Clostridia o__Clostridiales f__Clostridiaceae g__Clostridium s__celatum | Wean-finish | 0.99 | 0.02 ± 0.05 | 2.08 ± 1.08 | 0.00 |
| k__Bacteria p__Firmicutes c__Clostridia o__Clostridiales f__Clostridiaceae | Wean-finish | 0.97 | 0.01 ± 0.06 | 0.42 ± 0.11 | 0.00 |
| k__Archaea p__Euryarchaeota c__Methanobacteria o__Methanobacteriales f__Methanobacteriaceae g__Methanosphaera | Wean-finish | 0.96 | 0.02 ± 0.04 | 0.42 ± 0.22 | 0.00 |
| k__Bacteria p__Actinobacteria c__Coriobacteriia o__Coriobacteriales f__Coriobacteriaceae g__Collinsella s__aerofaciens | Wean-finish | 0.90 | 0.01 ± 0.03 | 0.12 ± 0.07 | 0.00 |
| k__Bacteria p__Firmicutes c__Clostridia o__Clostridiales f__Ruminococcaceae | Wean-finish | 0.89 | 0.02 ± 0.01 | 0.05 ± 0.05 | 0.01 |
| k__Bacteria p__Firmicutes c__Clostridia o__Clostridiales f__Clostridiaceae g__Clostridium s__celatum | Wean-finish | 0.88 | 0 ± 0 | 0.24 ± 0.20 | 0.00 |

(*Continued*)

**Table 6.** (Continued)

| | | | | | |
|---|---|---|---|---|---|
| k__Bacteria p__Firmicutes c__Clostridia o__Clostridiales f__Lachnospiraceae g__Coprococcus | Wean-finish | 0.82 | 0.01 ± 0.01 | 0.04 ± 0.03 | 0.00 |
| k__Bacteria p__Firmicutes c__Clostridia o__Clostridiales f__Lachnospiraceae g__Blautia | Wean-finish | 0.81 | 0.02 ± 0.02 | 0.09 ± 0.05 | 0.00 |
| k__Bacteria p__Firmicutes c__Clostridia o__Clostridiales f__Clostridiaceae | Wean-finish | 0.80 | 0.11 ± 0.30 | 0.46 ± 0.13 | 0.01 |

[1]Facility comparison done only with negative control treatment group.

[2]k represents kingdom, p represents phylum, c represents class, o represents order, f represents family, g represents genus, and s represents species. Bacteria are classified to be as specific as possible with the sequences available.

[3]Means expressed as percent relative abundance.

[4]*P*-value from Kruskal Wallis one-way analysis of variance.

In summary, the addition of subtherapeutic levels of chlortetracycline and sulfamethazine to nursery pig diets affected growth performance, metabolism, and gastrointestinal microbiome of weaned pigs. However, the experiment had a greater effect than antibiotic treatment on the pig metabolome, and the facility where the experiment was performed had a greater impact on the gut microbiome.

## Conclusions

These findings indicate that environmental characteristics of swine research facilities can have a significant effect on growth performance and the gut microbiome of pigs. In addition, studies are needed to investigate the environmental factors (i.e., temperature, humidity, microbial load), facility materials (i.e., steel, aluminum, plastic) and characteristics (i.e., age of facility, facility size, and ventilation system) of swine research facility that may affect the gut microbiome of pigs. A better understanding of how these environmental conditions impact the gut microbiome of pigs may provide an opportunity to have greater control of these variables in future experiments evaluating growth promoting feed additives in pig diets.

## Acknowledgments

The authors thank Margaret Schmidt, Jason Koch, and Kristy Segelhorst for assistance in maintaining and handling experimental animals, and data and tissue collection.

## Author Contributions

**Conceptualization:** Michaela P. Trudeau, Brenda de Rodas, Theodore P. Karnezos, Pedro E. Urriola, Andres Gomez, Milena Saqui-Salces, Chi Chen, Gerald C. Shurson.

**Data curation:** Michaela P. Trudeau, Huyen Tran.

**Formal analysis:** Michaela P. Trudeau, Wes Mosher, Huyen Tran, Andres Gomez, Chi Chen.

**Funding acquisition:** Brenda de Rodas, Theodore P. Karnezos, Gerald C. Shurson.

**Investigation:** Michaela P. Trudeau, Wes Mosher, Huyen Tran.

**Methodology:** Michaela P. Trudeau, Huyen Tran, Brenda de Rodas, Theodore P. Karnezos, Pedro E. Urriola, Andres Gomez, Milena Saqui-Salces, Chi Chen, Gerald C. Shurson.

**Project administration:** Brenda de Rodas, Gerald C. Shurson.

**Resources:** Brenda de Rodas, Pedro E. Urriola, Andres Gomez, Milena Saqui-Salces, Chi Chen.

**Supervision:** Brenda de Rodas, Gerald C. Shurson.

**Writing – original draft:** Michaela P. Trudeau, Pedro E. Urriola, Milena Saqui-Salces, Gerald C. Shurson.

**Writing – review & editing:** Michaela P. Trudeau, Huyen Tran, Brenda de Rodas, Theodore P. Karnezos, Pedro E. Urriola, Andres Gomez, Milena Saqui-Salces, Chi Chen, Gerald C. Shurson.

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
