## [Decision Letter · Decision Letter 0]

3 Feb 2023

PONE-D-23-00717Experimental facility had a greater effect on growth performance, gut microbiome, and metabolome in weaned pigs than feeding diets containing subtherapeutic levels of antibiotics: a case studyPLOS ONE

Dear Dr. Shurson,

Thank you for submitting your manuscript to PLOS ONE. After careful consideration, we feel that it has merit but does not fully meet PLOS ONE’s publication criteria as it currently stands. Therefore, we invite you to submit a revised version of the manuscript that addresses the points raised during the review process.

Both reviewers provide detailed comments and questions regarding components of the manuscript.  Specifically, both reviewers felt that some areas of the manuscript lacked the details required to fully explain either a comments, a method or to justify a conclusion.  It is mandatory that you respond to each and every comment made by the reviewers.in your revision and  cover letter.

We look forward to receiving your revised manuscript.

Kind regards,

Michael H. Kogut, Ph.D.

Academic Editor

PLOS ONE

Journal Requirements:

2. Partial research funding for conducting animal experiments was provided by Land O'Lakes/Purina and managed by BdR. Additional funding provided by Land O'Lakes/Purina to the University of Minnesota was managed by GCS for metabolomics and microbiome analysis and Research Assistantship support for MPT. BdR, TK, and HT from Land O' Lakes/Purina were involved in study design, data collection and analysis, decision to publish, and preparation of the manuscript". This information should be added to the COI/FD statement at a future check.

"Partial research funding for conducting animal experiments was provided by Land O' Lakes/Purina and managed by BdR. Additional funding provided by Land O' Lakes/Purina to the University of Minnesota was managed by GCS for metabolomics and microbiome analysis and Research Assistantship support for MPT. 

BdR, TK, and HT from Land O' Lakes/Purina were involved in study design, data collection and analysis, decision to publish, and preparation of the manuscript"

Reviewers' comments:

Reviewer's Responses to Questions

**Comments to the Author**

1. Is the manuscript technically sound, and do the data support the conclusions?

Reviewer #1: Yes

Reviewer #2: Yes

2. Has the statistical analysis been performed appropriately and rigorously? 

Reviewer #1: Yes

Reviewer #2: Yes

3. Have the authors made all data underlying the findings in their manuscript fully available?

Reviewer #1: No

Reviewer #2: No

4. Is the manuscript presented in an intelligible fashion and written in standard English?

Reviewer #1: Yes

Reviewer #2: Yes

5. Review Comments to the Author

Reviewer #1: In this paper, the authors address the effect of experimental facility and antibiotic feeding on the growth performance, microbiome, and metabolome of weanling piglets. While antibiotics are a known cause of enhanced animal growth performance, the mechanism remains unknown, so this paper is an important question in the field. The manuscript is clear and well-written with a thorough discussion. See below for detailed commentary:

-Data availability: States the data is fully available but not access information is provided. Apologies if I overlooked it, but please provide public accession numbers

-L25: "on" should be "in the" or something similar to fix grammar

-L79: I appreciate the listing of operating protocols, but unless there is something unusual about the SOPs at this location, this is not needed. Please keep the IACUC information, but the SRUs could be eliminated or more information should be provided in supplemental.

-L112-115: I greatly appreciate the honesty about infectious outbreaks in each experiment. It would be useful to know if S. suis was found in any of the microbiome samples from animals not eliminated from the studies. This could be altering their growth and microbiome (sub-clinically).

-DNA extraction: did you utilize any bead beating with this kit? If so, which type of bead? rpm? time?

-L151: There is much detail provided for the metabolomics, which is great, but was this done previously anywhere? If so, it should be cited. Or is this a modification of a previous protocol?

-L194: Which version of R was used?

-L195: which version of vegan?

-L216/221: one instance says "qiime2" and one says "Qiime 2". Please make consistent throughout the manuscript.

-L216: Which version of Qiime2?

-Tables 4 and on are very different formatting from tables 1-3. Please make consistent

-Figures 1, 2, and 4: For these ordinations, please provide % on each axis

-Figure 3 legend: please italicize p value

-Figures 1-4: It would be useful to include n values in the figure legends for reader clarity.

Reviewer #2: The topic is of relevance and the experimental design and methodology are sound. However, there are several issues that the authors should address. See below.

The statement about the databanke where the raw sequencing data were deposited is missing.

Abstract

Biomarker bacteria for AB treatment were identified. Those should be provided in the abstract.

Metabolites that were differently abundant across experiments/facilities/ages or for AB treatment should be provided in the results section of the abstract.

Also, dimension of changes for differently abundant parameters and growth performance should be provided in the abstract.

Introduction

The authors should elaborate more on how the investigation of the gut microbiome based on 16S rRNA sequencing and serum metabolome can help understanding the mechanisms how AGP work. The authors should provide the rationale for the selection of the gut sites and serum metabolomics.

Both the hypothesis and objective of the experiment needs improvement. Provide the gut sites for the microbiome and for the metabolome and also serum for the metabolome.

Results:

Regarding the presentation of data, there were three experiments with several growing phases. In addition, there was the AB treatment. Please organize the paragraphs in the results section accordingly. Otherwise, colleagues that do not carefully read the M&M section are lost.

Dimension of differences should be provided as % or fold-changes for all investigated paraemeters.

L247-250 Split sentence in 2 or 3 sentences and explain the observed effects separately for the different growing stages.

For metabolomics/microbiome, add the age of the pigs or the time postweaning when samples were collected.

L258 clear separation

Discussion

L311 Provide the length of the experimental phase postweaning.

L317 be more specific about the age-response; explain the mechanisms behind the observation that piglets grew differently when treated with AB.

When discussing the amino acid levels in serum, begin with explaining changes that may have occurred at gut level. Consider that AB could have upregulated the transporter expression at the gut mucosa. Check if amino acids that were differently abundant in serum utilize similar transporters. Also, consider an interaction of AB with the gut microbiome that due to an alteration in its composition may have interfered in amino acid uptake. Are there potential associations between metabolome effects and microbiome effects of the AB treatment.

L357-367 Re-write and shorten this paragraph. Changes in alpha diversity do not always occur. There were compositional changes which are more important to discuss.

Do not discuss data based on phylum level. Discuss data at family or genus level

Table and figures should be self-explanatory. Check that all abbreviations were defined at their first mentioning in the text. Provide the information whether data were across experiments and/or treatment groups in the table headings and figure legends.

Tables have different formats. Format tables according to the author guidelines.

6. PLOS authors have the option to publish the peer review history of their article (what does this mean?). If published, this will include your full peer review and any attached files.

Reviewer #1: **Yes: **Katie Lynn Summers

Reviewer #2: No

---

## [Author Response · Author response to Decision Letter 0]

1 Apr 2023

Reviewer #1: In this paper, the authors address the effect of experimental facility and antibiotic feeding on the growth performance, microbiome, and metabolome of weanling piglets. While antibiotics are a known cause of enhanced animal growth performance, the mechanism remains unknown, so this paper is an important question in the field. The manuscript is clear and well-written with a thorough discussion. See below for detailed commentary:

-Data availability: States the data is fully available but not access information is provided. Apologies if I overlooked it, but please provide public accession numbers

Response: Our apologies. The 16S rRNA sequence data analyzed in this manuscript are available in NCBI Sequence Read Archive under accession BioProject ID PRJNA950003. The link is: https://dataview.ncbi.nlm.nih.gov/object/PRJNA950003?reviewer=qc9rohdoa3u4huurtn063r90il

-L25: "on" should be "in the" or something similar to fix grammar

Response: Corrected as suggested.

-L79: I appreciate the listing of operating protocols, but unless there is something unusual about the SOPs at this location, this is not needed. Please keep the IACUC information, but the SRUs could be eliminated or more information should be provided in supplemental.

Response: Current line 94-97. The detailed information on SOPs was removed as suggested.

-L112-115: I greatly appreciate the honesty about infectious outbreaks in each experiment. It would be useful to know if S. suis was found in any of the microbiome samples from animals not eliminated from the studies. This could be altering their growth and microbiome (sub-clinically).

Response: In our 16s rRNA sequencing, S. suis was not identified in the analysis for any of the treatments even at low concentrations. Therefore, S. suis had negligible effects on growth and microbiome.

-DNA extraction: did you utilize any bead beating with this kit? If so, which type of bead? rpm? time?

Response: Yes, Bead beating was performed for 3 minutes at maximum speed (6m/s). This information was added to the methods section for the microbiome analysis (line 224-225)

-L151: There is much detail provided for the metabolomics, which is great, but was this done previously anywhere? If so, it should be cited. Or is this a modification of a previous protocol?

Response: Two references (20 and 21) were added into the method section of “Metabolomics analysis” to address this question. One on the general procedure (https://pubmed.ncbi.nlm.nih.gov/34096702/) (line 169), and the other on the HQ derivatization method, which was developed by our research group (https://pubmed.ncbi.nlm.nih.gov/24958262/). (line 182)

-L194: Which version of R was used?

Response: Rstudio version 3.5.3. was used and added to the manuscript on line 240.

-L195: which version of vegan?

Response: Vegan package 2.4.2. was used and added to the manuscript on line 241-242.

-L216/221: one instance says "qiime2" and one says "Qiime 2". Please make consistent throughout the manuscript.

Response: Corrected.

-L216: Which version of Qiime2?

Response: Version 2019.4. was used and added to line 236-237.

-Tables 4 and on are very different formatting from tables 1-3. Please make consistent

Response: The formatting of the tables is different because of the different types of analysis and data generated in this study. For example, Tables 1-3 provide averages from a targeted metabolomics analysis, while table 4 shows specific bacteria with significant indicator values. Only indicator values that were significant were presented, so there will not be a value for every treatment and every experiment. For this reason, the formatting had to be different to reflect the presentation of different types of data. No changes were made. 

-Figures 1, 2, and 4: For these ordinations, please provide % on each axis

Response: Figures 1, 2, and 4 were revised with the corrected axis labels

-Figure 3 legend: please italicize p value

Response: Corrected.

-Figures 1-4: It would be useful to include n values in the figure legends for reader clarity.

Response: n values were added to the legend in figures 1, 2, and 4.

Reviewer #2: The topic is of relevance and the experimental design and methodology are sound. However, there are several issues that the authors should address. See below.

The statement about the databank where the raw sequencing data were deposited is missing.

Response: This information is now provided. See previous comment for Reviewer 1. 

Abstract

Biomarker bacteria for AB treatment were identified. Those should be provided in the abstract.

Response: Bacteria families were added on lines 42-44.

Metabolites that were differently abundant across experiments/facilities/ages or for AB treatment should be provided in the results section of the abstract.

Response: Only 4 serum amino acids were biomarkers for the AB treatment, and a sentence was added to the abstract (line 44-45).

Also, dimension of changes for differently abundant parameters and growth performance should be provided in the abstract.

Response: We added that the change in growth performance was significantly different (line 38-39)

Introduction

The authors should elaborate more on how the investigation of the gut microbiome based on 16S rRNA sequencing and serum metabolome can help understanding the mechanisms how AGP work. The authors should provide the rationale for the selection of the gut sites and serum metabolomics.

Response: Additional explanation was added as suggested on lines 68-82.

Both the hypothesis and objective of the experiment needs improvement. Provide the gut sites for the microbiome and for the metabolome and also serum for the metabolome.

Response: Our hypothesis and objectives descriptions were revised on lines 86-91.

Results:

Regarding the presentation of data, there were three experiments with several growing phases. In addition, there was the AB treatment. Please organize the paragraphs in the results section accordingly. Otherwise, colleagues that do not carefully read the M&M section are lost.

Response: The results section was reorganized to present the antibiotic vs. negative control data in one subsection and the experiment 1, 2, and 3 comparison in a separate subsection to help improve clarity.

Dimension of differences should be provided as % or fold-changes for all investigated parameters.

Response: We respectfully disagree with this statement. The parameters investigated need to be analyzed and interpreted in the dimension of measurement. Interpretation of results are different when reporting data as a percentage change (constrained from 0-100) compared with quantification data such as animal growth rate (grams per day). No change.

L247-250 Split sentence in 2 or 3 sentences and explain the observed effects separately for the different growing stages.

Response: There were no significant differences in the other growing stages but we revised this sentence to clarify this on line 261-263.

For metabolomics/microbiome, add the age of the pigs or the time postweaning when samples were collected.

Response: We revised line 154-155 to indicate that pigs were 62 days of age when samples were collected.

L258 clear separation

Response: Corrected (line 278).

Discussion

L311 Provide the length of the experimental phase postweaning.

Response: Revised as suggested on line 335-336.

L317 be more specific about the age-response; explain the mechanisms behind the observation that piglets grew differently when treated with AB.

Response: Additional text was added as suggested on line 346-348.

When discussing the amino acid levels in serum, begin with explaining changes that may have occurred at gut level. Consider that AB could have upregulated the transporter expression at the gut mucosa. Check if amino acids that were differently abundant in serum utilize similar transporters. Also, consider an interaction of AB with the gut microbiome that due to an alteration in its composition may have interfered in amino acid uptake. Are there potential associations between metabolome effects and microbiome effects of the AB treatment.

Response: Thanks for this comment on potential mechanisms of observed changes in serum amino acids. We have since reviewed the literature on amino acid transporters (https://pubmed.ncbi.nlm.nih.gov/30177408/), which shows that the three AB-responsive amino acids do not share their uptake transporters. In addition, the concentrations of free amino acids in cecum digesta do not differ between AB and NC treatments, showing no clear influences from gut microbiota. Since lysine, phenylalanine, and histidine are essential dietary amino acids for pigs, we speculate that increased dietary intake in AB treatment might decrease the need of using essential amino acids for maintenance energy, leading to higher concentrations (which is described in our current discussion).

L357-367 Re-write and shorten this paragraph. Changes in alpha diversity do not always occur. There were compositional changes which are more important to discuss.

Response: We agree, and this paragraph was deleted.

Do not discuss data based on phylum level. Discuss data at family or genus level

Response: ok.

Table and figures should be self-explanatory. Check that all abbreviations were defined at their first mentioning in the text. Provide the information whether data were across experiments and/or treatment groups in the table headings and figure legends.

Response: Agreed. 

Tables have different formats. Format tables according to the author guidelines.

Response: See previous comment for Reviewer 1. The formatting of the tables is different because of the different types of analysis and data generated in this study. For example, Tables 1-3 provide averages from a targeted metabolomics analysis, while table 4 shows specific bacteria with significant indicator values. Only indicator values that were significant were presented, so there will not be a value for every treatment and every experiment. For this reason, the formatting had to be different to reflect the presentation of different types of data. No changes were made.

---

## [Editor Report · Decision Letter 1]

19 Apr 2023

Experimental facility had a greater effect on growth performance, gut microbiome, and metabolome in weaned pigs than feeding diets containing subtherapeutic levels of antibiotics: a case study

PONE-D-23-00717R1

Dear Dr. Shurson,

We’re pleased to inform you that your manuscript has been judged scientifically suitable for publication and will be formally accepted for publication once it meets all outstanding technical requirements.

Kind regards,

Michael H. Kogut, Ph.D.

Academic Editor

PLOS ONE

Additional Editor Comments (optional):

The revisions made to the manuscript and the responses of the authors' to the reviewers comments are acceptable.
---

## [Editor Report · Acceptance letter]

25 Apr 2023

PONE-D-23-00717R1 

Experimental facility had a greater effect on growth performance, gut microbiome, and metabolome in weaned pigs than feeding diets containing subtherapeutic levels of antibiotics: a case study 

Dear Dr. Shurson:

I'm pleased to inform you that your manuscript has been deemed suitable for publication in PLOS ONE. Congratulations! Your manuscript is now with our production department. 

Kind regards, 

on behalf of

Dr. Michael H. Kogut 

Academic Editor

PLOS ONE